# Presence of Mercury in an Arid Zone of Mexico: A Perspective Based on Biomonitoring of Mammals from Three Trophic Guilds

**DOI:** 10.3390/biology13100811

**Published:** 2024-10-11

**Authors:** Leticia Anaid Mora-Villa, Livia León-Paniagua, Rocío García-Martínez, Joaquín Arroyo-Cabrales

**Affiliations:** 1Museum of Zoology “Alfonso L. Herrera”, Faculty of Sciences, National Autonomous University of Mexico, Av. Ciudad Universitaria 3000, Coyoacán, Mexico City CP 04360, Mexico; psdanaid@live.com; 2Graduate Program in Biological Sciences, Graduate Studies Coordination, National Autonomous University of Mexico, Circuito de los Posgrados D-1, Coyoacán, Mexico City CP 04360, Mexico; 3Laboratory of Vegetal Ecology, National School of Biological Sciences, National Polytechnic Institute, Miguel Hidalgo, Mexico City CP 11340, Mexico; 4Atmosphere II Laboratory, Institute of Atmospheric Sciences and Climate Change, National Autonomous University of Mexico, Coyoacán, Mexico City CP 04510, Mexico; gmrocio@atmosfera.unam.mx; 5Laboratory of Archaeozoology, Subdirection of Laboratories and Technical Support, National Institute of Anthropology and History, Cuauhtémoc, Mexico City CP 06060, Mexico; arromatu@hotmail.com

**Keywords:** *Corynorhinus*, *Leptonycteris*, *Peromyscus*, xerophytic scrub, mercury, biomonitors

## Abstract

**Simple Summary:**

Mercury is one of the most common pollutants in agricultural and industrial areas; therefore, its study has become increasingly important. This element is often quantified using biomonitors, whose gradual response is representative of the entire community. This is especially important in areas where pollution is known or suspected, like the Mezquital Valley in central Mexico. Here, we aimed to determine the total mercury concentration in three species of small wild mammals that differ in diet to identify possible exposure threats in their habitat. We took hair and liver samples from two bat species and one wild mouse species over the course of a year. We found no difference in mercury concentration among species, but liver mercury was significantly lower during the dry season. This supports previous studies that pointed to wastewater as the main source of mercury exposure in this location. This is the first study to use small wild mammals as biomonitors in the Mezquital Valley, which is a critically contaminated site, as well as the first record of toxic metals in the protected bat species *Leptonycteris yerbabuenae*.

**Abstract:**

Mercury (Hg) has been extensively studied due to its impact on the environment and health, but its effects on wild mammal populations are still poorly known. Therefore, the use of biomonitors has gained importance. Our objective was to report and compare, for the first time, the amount of mercury in small mammals belonging to three trophic guilds and to provide an initial toxicology perspective in the Mezquital Valley, a critically polluted area of Central Mexico. We quantified total Hg from the hair and liver of a nectarivorous bat (*Leptonycteris yerbabuenae*), an insectivorous bat (*Corynorhinus townsendii*) and a granivorous mouse (*Peromyscus melanophrys*) using atomic absorption spectrometry during the dry and rainy seasons. We compared the mercury concentrations between seasons, species and matrices. In all species, the average mercury content was higher in hair than liver, and there was no correlation between matrices. There was no difference in mercury content among species. Hg concentrations in the livers of *P. melanophrys* and *C. townsendii* were lower during the dry season than the rainy season, suggesting a seasonal decline in mercury availability. All of the values detected were below the neurotoxicity threshold reported in small mammals (10 ppm); however, we propose constant monitoring of Hg in their environment and confirm the utility of these species as biomonitors.

## 1. Introduction

Mammals as a group have a wide geographic distribution, tend to have high relative abundances and occupy various trophic guilds, such as carnivores, nectarivores and frugivores. They also have long life cycles and reproductive stages that are easily distinguishable from each other [1,2]. Wild small mammals like rodents and bats are particularly useful models for assessing environmental metal pollution since their body size makes them easier to collect and handle than larger animals while still providing sufficient liver tissue, hair, etc. for toxicological studies [2]. These attributes give wild mammals strong potential as biomonitors—organisms that respond in a defined and quantifiable way to environmental changes [3] (pp. 139–159)—and whose responses reflect the effect of the disturbance on the rest of the community. Given their usefulness, biomonitors are increasingly common tools to evaluate the health of ecosystems and predict effects on human populations [4]. Rodents and bats are also excellent biomonitors because they frequently inhabit underground habitats such as caves, mines and burrows, which tend to be contaminated by metals. Also, studying wild small mammals from different trophic guilds provides us diverse perspectives of the local circulation of contaminants in plants, water and soil [1].

One of the most used matrixes in ecotoxicological research is the liver because it has an important role in metabolism, and specific hepatic enzymes transform toxic forms of many pollutants into less harmful molecules. Nonetheless, recent studies also rely on less invasive methods. One of the most relevant biological matrices under these criteria is hair, since it is easy to obtain and store. In addition, it can be taken from museum specimens and can be used for precise quantification of pollutants such as mercury [5].

Recently, the impact of human activities on mercury contamination in Mexico has been estimated using wild mammals as biomonitors; the approaches of these studies include neurophysiological, ethological and genetic analyses, the results of which offer novel and relevant perspectives [6,7]. Also, arid zones show greater seasonality than other ecosystems, such that the flow of contaminants and their effects on fauna differ significantly between the rainy and dry seasons [8]. Nonetheless, these exposure sites have been scarcely studied from an ecotoxicological perspective in Latin America [9]. Our study constitutes the first contribution of knowledge of mercury in wild mammals of the arid systems of central Mexico. This issue is relevant given the contamination from large mining, agriculture and industrial areas, where important populations of mammals are distributed and potentially exposed to pollutants [10]. Therefore, our main goal was to determine the amount of mercury in small mammals and understand the impact of factors such as seasonality and trophic guild on mercury concentration. We focused our study on mercury because of its ubiquity and its strong local impacts on the environment and health. Although important research has been published on mercury content and effects in mammals in general [11,12], our study area is still poorly known from an ecotoxicological perspective. This study will help to establish the bases for future conservation and environmental management plans.

## 2. Materials and Methods

### 2.1. Study Area

The Mezquital Valley is a hydrographic and cultural region located in the western part of the state of Hidalgo and is part of the Pánuco River basin (Figure 1). Its proximity to cinnabar deposits in the Sierra Gorda suggests that it receives sediments with high Hg content. The local climate is semi-arid with summer rains (BSkwg and BShwg), and the predominant vegetation comprises *Prosopis* sp., *Larrea* sp. and *Yucca filifera*, which form a mosaic of xeric scrub. One of the largest caves in the region is the Xoxafí grotto (coordinates: 20.388334 N, −99.027623 W, 2229 masl), which is a refuge for large colonies of nectarivorous and insectivorous bats [10]. Local agriculture is mostly represented by alfalfa, corn and oats, which together represent 83% of the total planted area. This agricultural system has resorted to the use of the largest system of urban wastewater reuse for irrigation in the world and uses pesticides, some of which contain mercury [13]. The Mezquital Valley also has intense industrial activity, such as the Tula oil refinery and five cement plants, one of which is in the vicinity of Xoxafi. It has been shown that metal contamination in this region has caused health problems in human populations, such as ophthalmic conditions and methemoglobinemia [14] (pp. 257–277). 

### 2.2. Study Species

*Leptonycteris yerbabuenae* is one of the largest nectar-eating bats in Mexico. It lives mainly in arid and semi-arid zones below 2600 masl. This species feeds on the nectar of agave plants and cacti, acting as a fundamental pollinator of these plants; this also makes *L. yerbabuenae* a highly economically important species for humans, as these plants are used to produce alcoholic beverages [15] (pp. 260–261). Due to its restricted habitat and its highly specialized diet, *L. yerbabuenae* is subject to special protection from the Mexican Government [16]. 

*Corynorhinus townsendii* is a small insectivorous bat that is distributed between sea level and 3160 masl. It feeds mainly on small, soft-bodied Lepidoptera, which are pests of agricultural importance [17]. *C. townsendii* mostly inhabits caves, mines and abandoned buildings.

*Peromyscus melanophrys* is a mouse species that is distributed in the arid and semi-arid zones of central Mexico, between 50 and 2750 masl. Its diet is based mainly on seeds, fruits and invertebrates [18] (pp. 753–754). It is therefore expected that any mercury detected will be from the soil, which may contain pesticides, as well as the items it feeds on. Previous studies suggest that there is a relationship between the underground habits of this species and the presence of heavy metals in its tissues, especially in mining areas of central Mexico [6,19].

### 2.3. Collection and Preparation of Samples

Liver and hair samples were taken from five adult males of each species during each sampling period. This sample size corresponds statistically to the minimum sample size needed to carry out the quantification [20]. Bats were caught using mist nets placed at the entrances to the Xoxafí grotto between dusk and midnight. Specimens of *P. melanophrys* were obtained on transects of Sherman-type traps, located in the bushes and rainfed crops within a radius of 1 km of the cave. Both *P. melanophrys* and *C. townsendii* were collected during both the rainy season (November 2021) and the dry season (March 2022). *Leptonycteris yerbabuenae* is migratory and is only present during the dry season. This bat species molts shortly after its arrival to Xoxafí, so the mercury contained in its hair at the time of capture reflects local exposure [21].

Specimens were taxonomically identified following Medellín et al. [22] (pp. 34–50) and Alvarez-Castañeda et al. [23] (pp. 154–155, 380–381) and their trophic guilds were defined according to the classification of Medellín [24] (pp. 334–354). We restricted our sampling to adult males because there is fluctuation in mercury storage in females associated with pregnancy and lactation [25]. The selected organisms were sacrificed by intraspinal injection of sodium pentobarbital, following the ethical treatment standards of the American Society of Mammalogists [26]. The liver was removed, and a sample of ventral hair was taken using dissecting forceps and scissors, as close to the root as possible. All dissection material was cleaned with 70% reagent-grade ethanol between each sample. All specimens were entered into the mammal collection of the Museum of Zoology of the Faculty of Sciences of the National Autonomous University of Mexico (MZFC-UNAM).

We quantified mercury in the liver and hair; they provide two different, potentially complementary, time periods of exposure. The Hg concentration in hair is proportional to the amount of Hg transported by the bloodstream to the hair follicle from the time the hair begins to grow until molting, which occurs twice a year and corresponds to the period before and after our sampling. Meanwhile, the Hg contained in the liver remains there for only a few days, after which it is mobilized [5,21]. Samples were preserved in vials and transferred to the UNAM Institute of Atmospheric Sciences and Climate Change (ICACC-UNAM). Next, 0.01 g of each sample was frozen, dried and then weighed into an individual Teflon container, using an analytical balance with a precision of 0.0001 g. All samples were subjected to an acid digestion process with HCl in a Mars-5 model microwave oven at 180 °C for 20 min to disintegrate the organic matter. The samples were allowed to cool for 24 h before opening the containers to avoid losses of volatile material. Then, KMnO_4_ was added as a reducing agent and sodium borohydride as a means of mercury hydride production.

Mercury analysis: The determination of total mercury (Hg) was carried out following the standardized method EPA 7470A Mercury in liquid residue (cold vapor technique). Internal quality controls and validation of the method were performed with the following parameters: The limit of detection (LOD = 0.158 ppb) was calculated from the formula $\bar{x}$ + 3 s and the limit of quantification (LOQ = 0.428 ppb) was obtained from $\bar{x}$ + 10 s, from the points of the calibration curves (n = 10). The linearity (r) was expressed by the average of the correlation coefficients of the calibration curves (r) Hg, r^2^ = 0.999. Sensitivity and precision were also evaluated in terms of repeatability and reproducibility. The concentrations of these curves were 0.5, 1, 2, 4, 6, 8 and 10 ppm. Accuracy assessment was calculated using the percentage of the relative standard deviation of repeatability and reproducibility (Appendix A). After every eight readings, one point on the curve was measured again to corroborate the veracity of the results. All concentrations are reported in ppb on a dry basis. To achieve more accurate measurements, all samples were analyzed in duplicate. Reference material was used for Hg analysis (standard is traceable to NIST SRM 3133 [27], ISO 9001:2015 [28] certified and ISO/IEC 17025:2017 [29] and ISO 17034:2016 [30] accredited).

All calibration curves and samples were analyzed in an atomic absorption spectrometer (EAA GBC 932 AA. GBC, Keysborough, Victoria, Australia), using the cold vapor technique with a hydride generator (GBC HG 300C, GBC, Keysborough, Victoria, Australia). Blanks were analyzed for each batch of samples in the same equipment. Stock solutions for the calibration curves were prepared daily.

Statistical analysis: We performed all analyses in the R programming language, version 4.0.3 [31]. We first calculated the basic elements of descriptive statistics, then evaluated normality within each species using the Shapiro–Wilk test (*C. townsendii*: *p* = 0.071; *P. melanophrys*: *p* = 0.051 and *L. yerbabuenae*: *p* = 0.461) and the homoscedasticity among the three species using Bartlett’s test (*p* = 0.866). Since both normality and homoscedasticity were confirmed, we used parametric analyses. We used Student’s *t* tests for independent samples to compare the values between seasons and between matrices within each of the species and among all samples. A one-way ANOVA was used to detect significant differences in mercury concentration among species. A Pearson correlation test was used to assess the relationship between hair and liver mercury concentration within each individual. Finally, we performed a two-way ANOVA to evaluate the effect of the matrix and the species on the concentrations. Tukey post hoc tests were used to determine pairwise differences when ANOVAs were significant. All statistical analyses considered a significance threshold (α) of 0.05.

## 3. Results

The mercury concentrations in the tissues of the three species ranged from 1.188 ppb (in a *P. melanophrys* liver during the rainy season) to 12.499 ppb (in the hair of a *C. townsendii* specimen during the rainy season). All specimens had mercury levels that were above the detection limit (Table 1). All of the values in this study were below 10,000 ppb, which corresponds to the minimum concentration reported as causing neuromotor disorders in other bat species [12,32].

The general comparison among the three species showed that although the highest values were recorded in *C. townsendii* (Figure 2), the mean concentration of mercury did not differ significantly among taxa (*p* = 0.528). When comparing between the two matrices using all species and seasons, we found significantly higher mercury concentrations in hair than in liver (*p* = 8 × 10^−8^). When comparing hair mercury concentration among the three species, although the highest mean contents were recorded in the hair of *C. townsendii* (an insectivore), followed by that of *L. yerbabuenae* and *P. melanophrys* (primary consumers), the difference among species was not significant (*p* = 0.341), consistent with the test considering grouping hair and liver values. Similarly, there was no significant difference among the three species in liver mercury (*p* = 0.492).

When comparing between matrices within each species, we found that the concentration of mercury in the hair was significantly higher than in the liver (*C. townsendii*, *p* = 0.0008; *L. yerbabuenae*, *p* = 0.0001 and *P. melanophrys*, *p* = 0.009). When considering the sampling season (for *C. townsendii* and *P. melanophrys,* which were sampled during both seasons), we found that Hg concentration was significantly higher during the dry season (*p* = 0.038) than the rainy season. When compared within each species and matrix, we found significantly higher mean mercury concentration during the dry season in the liver of *C. townsendii* (Student’s t, *p* = 0.003), and of *P. melanophrys* (*p* = 0.0001) (Figure 3).

The correlation between the amount of mercury in the liver and in the hair of all the specimens in the study is weak (Pearson coefficient R = 0.43). However, performing the correlation test for each species separately showed that while the hair and liver values are weakly correlated in *C. townsendii* (R = 0.043) and *P. melanophrys* (R = 0.533), they are strongly correlated in *L. yerbabuenae* (R = 0.915). The ANOVA result shows that the matrix in which mercury is quantified exerts a significant effect on the amount recorded (*p* = 1.6 × 10^−5^). On the other hand, neither the species nor the interaction between the factors exert a significant effect on the amount of mercury present in the system.

## 4. Discussion

The validity of the use of the species in this study as biomonitors coincides with Ramos-H et al. [7], who showed that assemblages of bats are excellent biomonitors of environmental quality in highly polluted areas, such as Mexico City. In our study area, Pérez et al. [13] and Guédron et al. [33] established that the concentration of mercury in irrigation water (2 to 5 ppb) exceeds the maximum permissible limits in Mexican legislation (1 ppb). They demonstrated that the main source of mercury in the Mezquital Valley is water and that there is a positive correlation between the time of irrigation and the concentration of heavy metals in plant tissues. Recent studies [34] (pp. 215–231) and [35] confirm that the main source of mercury in this locality is wastewater, and maximum concentrations in irrigation water can be as high as 4500 ppb ($\bar{x}$ = 3800 ppb).

Total concentrations in mammalian hair and liver can vary by orders of magnitude depending on factors including diet and habitat. In the liver, metallothioneins bind to mercury and reduce its toxic effect, making the liver one of the main structures responsible for the metabolism of mercury. The half-life of contaminants in the liver can reach just a few days [36]. On the other hand, hair constitutes a long-term elimination route since mercury accumulates in the hair fiber as it grows. As such, mercury concentration in hair continues to accumulate over longer periods of time and therefore tends to be higher than in liver. Previous studies have shown a significant correlation between the mercury concentration in hair and organs such as the liver and the brain, and the Hg concentration in hair has also been correlated with its concentrations in environmental matrices [37]. 

The values found in this study are similar to those recorded in the livers of insectivorous bats in tropical agroecosystems subjected to pesticide application [38]. However, unlike this work, other authors have found differences between trophic guilds; for example, insectivorous bats have been shown to have higher concentrations than nectarivorous bats [39]. In granivorous rodents, the amount of Hg in hair reaches up to 380,000 ppb in mining areas [6]. Nevertheless, our average was 6.85 ppb, which indicates that, despite local pollution, mammals in Xoxafí are subject to less mercury incorporation than in areas with intensive mining exploitation. 

Herbivores, such as *L. yerbabuenae* and *P. melanophrys,* come into contact with mercury through the ingestion of contaminated vegetation [39]. Plants accumulate a higher concentration of pollutants in the leaves, stems, and roots, and the consumption of these parts is riskier than others, such as fruits and seeds [40]. On the other hand, insectivory comprises prey from multiple taxonomic groups, trophic levels, and habitats. Therefore, insects and their predators are exposed to Hg in different chemical forms and from multiple sources [41]. Some insectivorous bats can even consume up to 50% of their body weight in insects per night, which represents several tons of arthropods, that play an important role in the mobilization of metals [11,42].

Adachi et al. [43] point out that once Hg enters the body, the concentration in the organism depends on protein intake and glutathione metabolism, which is why carnivorous and insectivorous organisms are significantly more exposed and are more prone to accumulate larger amounts of this element than other trophic guilds. Considering this, we expected the insectivores to be the most vulnerable group. However, no significant difference was recorded between the guilds. This may be due to the arid conditions of the area. Hg in the soil of arid systems tends to evaporate with water during the dry season and enters plants through their stomata. Therefore, it is available mostly in their aerial parts. It also becomes concentrated in the soil during most of the dry season [44]. During the rainy season, in contrast, vegetation in arid environments absorbs and accumulates more Hg in structures such as roots [45]. Under that scenario, the Hg present is less available for ingestion by nectarivorous, insectivorous and frugivorous organisms. This is consistent with the low liver concentrations found in *C. townsendii* and *P. melanophrys* during the rainy season. The decrease in the amount of Hg in the liver during the rainy season also suggests an atmospheric origin of this pollutant, since the winds coming from the Sierra Gorda contain particulate mercury, which is dragged towards the Mezquital Valley and deposited by rain in the habitat during the rainy season [33]. Nonetheless, quantitative studies are required to describe the importance of this phenomenon.

In addition to diet, the habitat defines exposure to pollutants. For this reason, the preference of certain mammals for underground burrows makes them more likely to be in contact with Hg in most of its chemical forms [46]. This is especially important in the case of fossorial mammals. On the other hand, cave bats are a special case, since in addition to being exposed to the contaminant itself, they live under unique environmental conditions, such as accumulation of waste, high population densities and high temperatures. This makes them more susceptible to infectious diseases, which increase in the presence of pollutants such as Hg [7]. Therefore, mines and caves can constitute sources of damage from metal exposure and should be broadly monitored [6].

This is the first study on mercury accumulation in *L. yerbabuenae*, which had similar concentrations to those of *C. townsendii* and *P. melanophrys*. Despite having a broad home range during its migration throughout Mexico and the United States [15], the Hg concentration quantified in this study corresponds only to the dry season, during which this bat species inhabits the Mezquital Valley. Also, *L. yerbabuenae* shows a high correlation between the average mercury content in hair and liver, confirming that hair can provide similar information on Hg accumulation as liver without requiring sacrificing individuals. This, together with its status as a protected species, makes it an interesting species for future non-invasive studies. Future studies should also compare *L. yerbabuenae* Hg concentrations with readings from other resident species, as well as expand data collection from *L. yerbabuenae* along its migratory route.

The relationship between the concentration of mercury in biomonitors and the environment is not linear. Hernout et al. [5] showed that the concentration of mercury in the hair of bats does not directly correlate with the total amount of environmental pollutants, but it is an indicator of the bioavailable fraction of metals in the soil. Likewise, atmospheric processes are strongly related to the availability of this element to living beings [8]. This demonstrates the need for interdisciplinary studies that describe the dynamics of metals in the atmosphere, water and soil, together with their storage in living beings. 

Finally, considering the risk of contamination currently faced by wild populations, it is extremely important to develop studies that provide baseline values of the amount of contaminants present in mammals. For this reason, we propose that investigators prioritize habitats such as caves with colonies of insectivorous bats, systems subjected to the use of agrochemicals, and the use of models for non-invasive studies.

## 5. Conclusions

This study confirms the presence of Hg in the hair and liver of rodents and bats from Xoxafí Cave and constitutes the first contribution to knowledge of Hg in the arid systems of the Mezquital Valley. We conclude that hair is an adequate matrix for mercury quantification, particularly in protected species such as *L. yerbabuenae*, since this technique is minimally invasive and it complements studies in internal matrices such as the liver. The presence of higher amounts of Hg in hair than in liver in all trophic guilds analyzed suggests exogenous accumulation from sources other than food or inhalation. Additionally, the increase in hepatic mercury during the dry season shows that the availability of this element is seasonal, which emphasizes wastewater as the main local source of Hg. The average concentration of mercury in the species analyzed is not above acute neurotoxic values; however, it is advisable to carry out constant monitoring of the ecosystem’s health, preferably using non-invasive techniques. 

## Figures and Tables

**Figure 1 biology-13-00811-f001:**
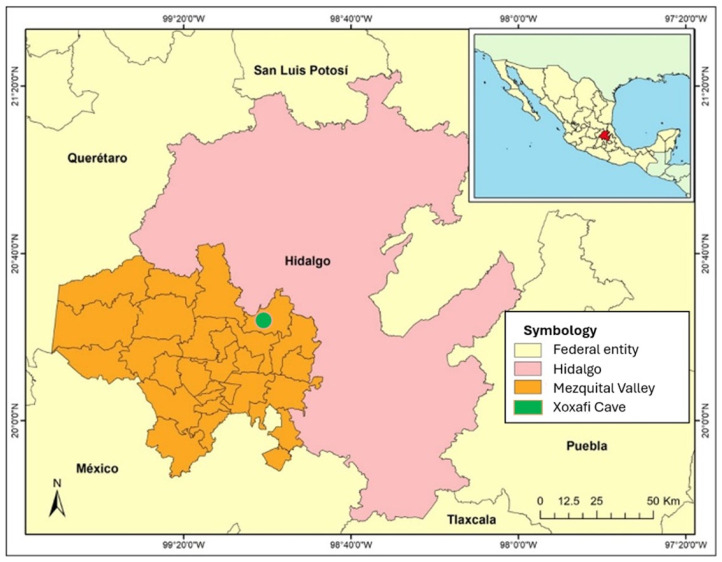
Geographic location of the study area.

**Figure 2 biology-13-00811-f002:**
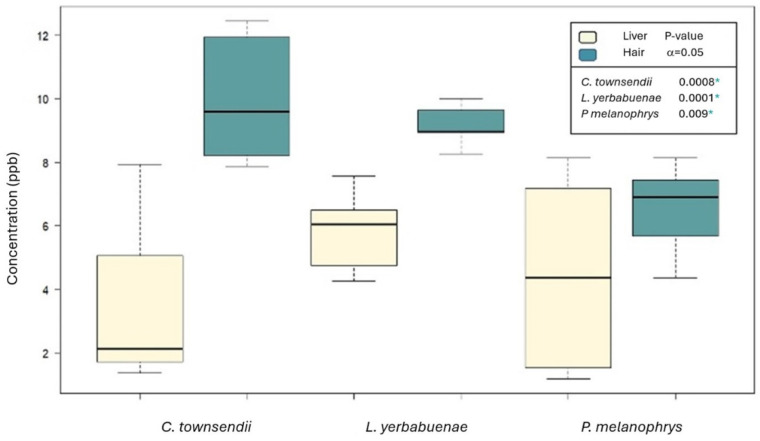
Comparison of mean mercury concentrations between matrices and species (n = 25 per matrix, in ppb on basis of dry weight). * Significant difference between means (α = 0.05).

**Figure 3 biology-13-00811-f003:**
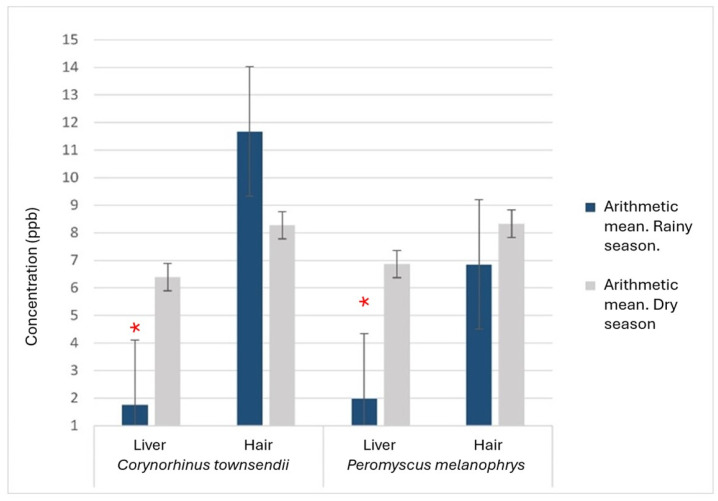
Comparison of mean mercury concentrations between seasons and species (n = 20 per species. Dry weight). * Significant difference between means (α = 0.05).

**Table 1 biology-13-00811-t001:** Summary of descriptive statistics of the concentrations (ppb) of mercury quantified in the present study (Nt = 50. Dry weight).

	*C. townsendii*				*L. yerbabuenae*				*P. melanophrys*			
	Liver		Hair		Liver		Hair		Liver		Hair	
	Rainy	Dry	Rainy	Dry	Rainy	Dry	Rainy	Dry	Rainy	Dry	Rainy	Dry
$\bar{x}$	1.756	6.382	11.672	8.275	-	5.823	-	9.156	1.975	6.865	6.844	8.327
Máx.	2.125	7.925	12.438	8.803	-	7.567	-	9.995	4.219	8.125	10.941	11.081
Mín.	1.375	4.943	10.375	7.875	-	4.266	-	8.264	1.188	4.353	5.033	7.277
C.I. (95%)	(1.398–2.114)	(3.839–8.926)	(10.209–13.135)	(7.659–8.891)	-	(4.169–7.477)	-	(8.317–9.995)	(0.405–3.544)	(5.477–8.253)	(3.831–9.857)	(6.841–9.813)
S	0.288	1.598	0.919	0.387	-	1.332	-	0.675	1.263	1.322	1.085	1.415

## Data Availability

The original contributions presented in the study are included in the article/online Appendix A. Further inquiries can be directed to the corresponding author(s).

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
