# Peer review of "Presence of Mercury in an Arid Zone of Mexico: A Perspective Based on Biomonitoring of Mammals from Three Trophic Guilds"

_biology, 2024, doi:10.3390/biology13100811_

Round 1

Reviewer 1 Report

Comments and Suggestions for Authors

General comments

Only one heavy metal, a small number of samples, within only two seasons are not enough to be able to draw conclusions. It is local study, and it can be taken as preliminary work. Why did authors choose to analyze only mercury? I am afraid that this is not enough for paper being worth publishing.

Specific comments

Line 52: additional for” should be deleted

Line 56-58: The sentence should be rewritten.

Line 74-75: The sentence should be rewritten.

Line 78: The reference is missing.

Line 80-96: This part should be rewritten to better reflect the studied area and the aim of the study, in order to be in good connection with the previous part of the introduction section. Authors should better justify why only mercury was measured among different heavy metals. The health adverse effects related to mercury exposure should be mentioned as well. See recently published papers for more details (https://doi.org/10.1186%2Fs12940-023-01029-z https://doi.org/10.3390%2Ftoxics12070490 https://doi.org/10.1007%2Fs10646-024-02747-x )

Materials and Methods: The sample preparation method and the method performance results should be better explained in more details (such as LoD and LoQ, accuracy, precision) in order to give evidence of the suitability of the methodological approach. It is not clear which certified reference material was used. Moreover, the total number of samples is not clear. How much per season?

Discussion and Conclusion: It is not clearly understood how exactly the results of this study can have wider impact

Comments on the Quality of English Language

Minor editing is needed.

Reviewer 2 Report

Comments and Suggestions for Authors

A brief summary

The aim of presented paper was to compare mercury concentration in hair and liver of two bat species and one mouse species, and to discern if sampling season (dry/rainy) affects concentration of this element. The research is scientifically sound, includes species with different diet preferences, includes seasonal aspect, and, generally, contributes to ecotoxicological knowledge on small mammals.

General concept comments

Article: The main concept for this paper is well thought out, and the described methodology, in detail, is complete and correct. The hypothesis is easily tested and reproduced.

Review: The paper has no discernible gaps in either general idea or methodological approach. Generally, the paper is well written, easy to understand, and contributes to scientific knowledge on ecotoxicology of small mammals (specifically – bats).

Specific comments

Introduction

Lines 48-51: Different sources should be cited. The authors cited  Jones at al 2009. for sentences about mammals in generals and this publication focuses specifically on bats.

Lines 53-54: Sentence about captivity and management is redundant and could be removed from manuscript.

Lines 69-70: The sentence „Also, because liver is one of the main target organs of pollutants as toxic metals“ seems either incomplete or rewritten because in current form it is very hard to understand what authors wanted to state.

Lines 80-83: This part should be ending of the Introduction section and further expanded with aims of the research.

Lines 84-96: This entire paragraph should be placed in Materials and methods.

Materials and methods

Figure 1: Figure needs to be changed because it does not include actual sampling location (and its surroundings).

Line 87: It is unclear what (BSkwg and BShwg) is. Does it have meaning or is it something authors forgot to remove from the manuscript during writing?

Lines 103-104: Authors thoughts about contamination source for this species should be removed from Materials and methods and implemented in Discussion.

Lines 107-108: Same comment as above.

Lines 111-115. Same comment as above.

Lines 130-133: This sentence should ne removed since these additional measurements (such as weight) have no impact on the research or manuscript.

Lines 138-142: Authors need to cite sources for these statements (about Hg concentration in hair and liver).

Additional comment for Materials and methods: authors should state if obtained results are presented as dry weight or wet weight (this should be done also in the Figure 2 and 3 and Table 1).

Results

Figure 3. n = 25 should be changes to n = 20 since authors stated that they sampled 5 individuals per species per season.

In text, the authors state that they sampled „hair“, and in Figures 2 and 3, as well as Table 1, the used term is „fur“. This needs to be uniformed.

Discussion

Lines 222-226: These sentences should be changed since (as is currently written) the reader could imply that Hg is connected to water, irrigation and plants in Mezquital Valley. However, paper by Zambrano-Garcia et al. (2009), which is cited, does not include mercury in research.

Lines 253-255: Cited paper by Currie et al. (1997) state that insectivorous bats can eat up to 1 g of insects each night and not 50% of the body weight as authors wrote in this paper. This needs to be changed.

Comments on the Quality of English Language

The quality of English language could be improved in some sentences.

Reviewer 3 Report

Comments and Suggestions for Authors

1-      Divide the manuscript into clear parts (introduction, methods, results and discussion followed by conclusion)

2-      Double check the English language of the manuscript

3-      Add the IRB and ethical consideration in the manuscript

4-      Insert each figure and table below the paragraph describing them

5-      Remove the appendix and add it in a supplementary file

6-      Write the references according to MDPI style format

7-      Why is the mercury concentrated mostly in the hair?

8-      In ‘Author Contributions’ part, write the initials of the authors and not full names

9- did you analyse the amount of mercury in the plants of this area?

Comments on the Quality of English Language

minor revisions are needed

Reviewer 4 Report

Comments and Suggestions for Authors

Abstract should be rewritten as it does not give the "overall picture". It should provide information about: introduction/background, purpose/objectives, methdology, results and concluions.

Round 2

Reviewer 1 Report

Comments and Suggestions for Authors

General comments:

Although authors tried to improve the paper several issues are still presented. The novelty and the necessity of this type of research should be better reflected in the introduction section. I encourage authors to give more efforts in rewriting the introduction section and material and methods.

Specific comments:

Line 35: Authors did not use several techniques, only one for quantification

Line 35-36: This part of sentence should be deleted „considering that the main contribution of mercury in the area is wastewater from urban areas, that reaches the soil and crops.“

Line 59: The sentence should be rewritten in a proper way. „As if“ is not proper phrase

Introduction section: Authors should better justify why only mercury was measured among different heavy metals that are present based on the mentioned paper in the studied area (arsenic, lead).

Mercury analysis: The authors did not solve this issue in a proper way. The sample preparation method and the method performance results should be better explained. Please try to organize sentences in a better way. How LOD and LOQ values are calculated should be presented in the paper as well as the obtained values.

Comments on the Quality of English Language

With the help of English professional translator, the manuscripts should be checked and improved. 

Reviewer 3 Report

Comments and Suggestions for Authors

The authors have addressed all the required modifications and the manuscript can be accepted 
